# Teledermoscopy in the Diagnosis of Melanocytic and Non-Melanocytic Skin Lesions: Nurugo^TM^ Derma Smartphone Microscope as a Possible New Tool in Daily Clinical Practice

**DOI:** 10.3390/diagnostics12061371

**Published:** 2022-06-02

**Authors:** Federica Veronese, Vanessa Tarantino, Elisa Zavattaro, Francesca Biacchi, Chiara Airoldi, Massimo Salvi, Silvia Seoni, Francesco Branciforti, Kristen M. Meiburger, Paola Savoia

**Affiliations:** 1SCDU Dermatologia, AOU Maggiore della Carità, C.so Mazzini 18, 28100 Novara, Italy; federica.veronese@med.uniupo.it (F.V.); vanessa.tarantino30@gmail.com (V.T.); 2Department of Translational Medicine, University of Eastern Piedmont, Via Solaroli 17, 28100 Novara, Italy; chiara.airoldi@uniupo.it; 3School of Medicine, University of Eastern Piedmont, Via Solaroli 17, 28100 Novara, Italy; 20017745@studenti.uniupo.it; 4Biolab, Polito^BIO^Med Lab, Department of Electronics and Telecommunications, Politecnico di Torino, 10129 Torino, Italy; massimo.salvi@polito.it (M.S.); silvia.seoni@polito.it (S.S.); francesco.branciforti@polito.it (F.B.); kristen.meiburger@polito.it (K.M.M.); 5Department of Health Science, University of Eastern Piedmont, Via Solaroli 17, 28100 Novara, Italy; paola.savoia@med.uniupo.it

**Keywords:** teledermatology, teledermoscopy, skin cancer diagnosis

## Abstract

**Background:** Due to the COVID-19 pandemic, teledermoscopy has been increasingly used in the remote diagnosis of skin cancers. In a study conducted in 2020, we demonstrated a potential role of an inexpensive device (Nurugo^TM^ Derma) as a first triage to select the skin lesions that require a face-to-face consultation with dermatologists. Herein, we report the results of a novel study that aimed to better investigate the performance of Nurugo^TM^. **Objectives:** (i) verify whether the Nurugo^TM^ can be a communication tool between the general practitioner (GP) and dermatologist in the first assessment of skin lesions, (ii) analyze the degree of diagnostic–therapeutic agreement between dermatologists, (iii) estimate the number of potentially serious diagnostic errors. **Methods**: One hundred and forty-four images of skin lesions were collected at the Dermatology Outpatient Clinic in Novara using a conventional dermatoscope (instrument F), the Nurugo^TM^ (instrument N), and the latter with the interposition of a laboratory slide (instrument V). The images were evaluated in-blind by four dermatologists, and each was asked to make a diagnosis and to specify a possible treatment. **Results:** Our data show that F gave higher agreement values for all dermatologists, concerning the real clinical diagnosis. Nevertheless, a medium/moderate agreement value was obtained also for N and V instruments and that can be considered encouraging and indicate that all examined tools can potentially be used for the first screening of skin lesions. The total amount of misclassified lesions was limited (especially with the V tool), with up to nine malignant lesions wrongly classified as benign. **Conclusions:** Nurugo^TM^, with adequate training, can be used to build a specific support network between GP and dermatologist or between dermatologists. Furthermore, its use could be extended to the diagnosis and follow-up of other skin diseases, especially for frail patients in emergencies, such as the current pandemic context.

## 1. Introduction

Since 1995, when it was applied for the first time, teledermatology (TD) has been gradually applied in almost all dermatology sub-specialties [1,2] and, in the last few years, has also been proposed in the dermatological research context [3,4,5]. TD finds its natural application in the diagnosis of melanocytic and non-melanocytic skin cancer with its specific branch, namely teledermoscopy (TDS). Since 2010, many private websites, such as *Dermoscopedia* (from the International Dermoscopy Society), the ISIC (International Skin Imaging Collaboration) archive, and the iDS score project [1,6], have been built with the specific aim to collect, catalog, and share dermoscopic case libraries for educational purposes. In addition, numerous algorithms have been developed for automatic dermoscopic image segmentation and/or classification, with a recent focus on deep learning models, i.e., the convolutional neural networks (CNNs) [7,8,9,10,11].

In 2020, we conducted a study to verify the diagnostic reliability of images acquired through an inexpensive smartphone microscope (Nurugo^TM^), employing CNNs to classify malignant melanoma (MM), melanocytic nevus (MN), and seborrheic keratosis (SK). The CNN was tested on three different test sets: (i) ISIC images, (ii) images acquired with the Nurugo^TM^, and (iii) images acquired with a conventional dermatoscope [12]. The Nurugo^TM^ device showed an accuracy comparable to conventional dermatoscope (up to 80%), which demonstrates a potential role performing the first triage in TDS, selecting the skin lesions that require a face-to-face consultation with dermatologists [12].

Based on these preliminary results, herein we report a novel TDS study, intending to verify whether the Nurugo^TM^ Derma can be useful to evaluate lesions through TD, in two-way communication between the general practitioner (GP) and dermatologist. To meet this goal, we evaluated the agreement degree between four dermatologists with different dermoscopy experiences in evaluating images of skin lesions acquired with three different tools. The secondary outcomes were: (i) to analyze the degree of diagnostic–therapeutic agreement between the four dermatologists and (ii) to consider the number of potentially serious diagnostic errors, i.e., the number of malignant lesions misclassified as benign.

## 2. Materials and Methods

### 2.1. Images Collection

One hundred and forty-four skin lesion images were collected. Based on clinical characteristics, the images were classified as follows: 75 melanocytic naevi (MN), 6 melanomas (MM), 27 basal cell carcinomas (BCC), and 36 seborrheic keratoses (SK). For all lesions, images were acquired with three different devices: (i) a portable dermatoscope (Figure 1A) connected to a reflex camera (Figure 1B), (ii) the Nurugo^TM^ Derma (Figure 1G) connected to a smartphone (Figure 1H), and (iii) Nurugo^TM^ Derma connected to the smartphone (Figure 1I) with the interposition of a laboratory slide. The images were acquired by an MD student, adequately trained in the use of these devices by an experienced dermatologist. We assimilated the student to the GP, due to the limited experience in recognizing skin lesions. The images were acquired during routine visits, after collection of signed informed consent from the patients; the study was approved by the Local Ethics Committee (protocol 173/18, approved on 12 December 2018). To ensure the protection of the patients’ data, images were encoded with an alphanumeric code. All the patients included in the study were treated based on the clinical diagnosis made by the expert dermatologist with the help of the conventional dermoscopy—in detail, all suspected lesions were surgically excised and histologically examined, whereas patients with lesions evaluated as clinically benign continued the regular dermatological follow-up.

### 2.2. Images Pre-Processing

The images collected with the portable dermatoscope and reflex camera, after coding, were sent to the Biolab engineers (Polito^BIO^Med Lab, Department of Electronics and Telecommunications) for optimized cropping and color normalization according to specific ad hoc algorithms. Indeed, cropping allows the reduction of the large area of black pixels around the central circular area containing the photographed lesion, also making the aspect ratio square. The employed algorithm proceeds according to the following steps:Transformation of the image into greyscale and binarization through global thresholding (Figure 1C,D);Identification of the coordinates and dimensions of the central area (Figure 1E);Cropping (Figure 1F).

The same images were also acquired with the smartphone assembled with the Nurugo^TM^ Derma lens and related to the respective dermatoscopic images. The lesions were acquired both in “normal” and in “epiluminescence” settings, i.e., after applying paraffin oil and positioning a laboratory slide as an interface between the skin and the Nurugo. Due to the presence of an artifact that limits the field of view (FOV) when an epiluminescence image is acquired, for large lesions, more images were acquired by modifying the positioning of the Nurugo and the slide. To circumscribe the portion without reflection artifacts, an automatic segmentation algorithm based on the circular Hough transform was developed and applied (Figure 1J–L).

### 2.3. Images Classification

We collected 144 images that were adequately codified and normalized and/or cropped, and were then entered on the REDCap (Research Electronic Data Capture) application and divided based on the acquisition tool, with an encoding that did not allow users to trace the specific device used, and in a different sequence to randomize the evaluation array and avoid bias.

In particular:-Instrument F (dermatoscopic images): ordered from 1 to 144;-Instrument N (Nurugo^TM^ images): ordered from 144 to 1;-Instrument V (Nurugo^TM^ images in epiluminescence): ordered from 72 to 144 and from 1 to 71.

Four identical records were created—each one was assigned to a skilled dermatologist who evaluated the images and expressed both a diagnostic and therapeutic opinion, through a multiple-choice questionnaire. No additional clinical and/or anamnestic data were available to the dermatologists. For each image, the dermatologist chose a diagnostic option from those proposed: (i) MN, (ii) MM, (iii) BCC, or (iv) SK; in addition, the option “yes” or “no” indicated the potential recommendation for surgical removal. For a better interpretation, see Figure 2.

The dermatologists involved in the study had different experience in the dermoscopic field (dermatologists 1 and 3: 5–10 years; dermatologists 2 and 4: >10 years).

### 2.4. Statistical Analysis

Based on literature [13] and our previous experience, we assumed that comparison between a classic dermatoscope and other different dermatological tools had a disagreement of 0.28 with a Cohen’s kappa of 0.61. Considering an interval confidence of 0.95 with a width of 0.30, a minimum of 135 skin lesions were required. To avoid problems related to missing data, blurry photos, or others, we performed 144 observations.

The first step was to evaluate the agreement between dermatologists on the real clinical diagnosis (i.e., diagnosis made through conventional dermoscopy and used as a standard reference, or, in case of excised lesions, the histological report); diagnosis and possible indications for surgical removal have been remotely evaluated (TDS) and the agreement between dermatologists was noticed.

Descriptive analyses were also conducted: for each instrument used (F, N, V) and for each of the four dermatologists involved, the absolute and relative frequencies of each diagnosed lesion and the relative decision about its treatment were calculated.

The agreement degree between the real clinical diagnosis (C), used as a standard reference, and the diagnosis related to each instrument (primary outcome) was calculated using the Cohen’s kappa with a 95% confidence interval [95% CI]. The Landis and Koch scale was used as a reference system [14].

The agreement between dermatologists was then calculated concerning the three tools used, both in terms of diagnosis and of treatment recommendations, using the Fleiss kappa with 95% CI, a statistical parameter that allows for evaluation of the degree of agreement between several elements [15]. The reference values of this parameter are the same as Cohen’s K.

Finally, the “serious diagnostic errors” (that is, the number of malignant lesions [MM and/or BCC] misclassified as benign [Mn and/or SK]), were calculated. This value was expressed as % of the total malignant lesions analyzed.

All statistical analyses were conducted using SAS 9 and R.

## 3. Results

Table 1 summarizes the diagnoses suggested by the four dermatologists for each lesion/instrument (dermatoscope F, Nurugo N, Nurugo in epiluminescence setting with slide V), together with the real clinical diagnosis (C), and reports the final clinical decision (removal or not).

A concordant diagnostic approach was found among dermatologists 1, 2, and 3, with a tendency to overestimate melanomas with instruments N and V. Despite the 6 real clinical diagnoses, 7 lesions were classified as MM with instrument N and 17 with the instrument V for dermatologist 1, 16 with N and 29 with V for dermatologist 2, and 24 with N and 20 with V for dermatologist 3. Dermatologist 4 was better aligned to the real clinical condition for the images evaluated with instruments F and N, and classified 72 MN, 3 MM, 41 SK, and 28 BCC with instrument F and, respectively, 46, 3, 57, and 38 with instrument N. However, with instrument V, dermatologist 4 overestimated the SK and underestimated the MN (respectively, 64 compared to 36 effective (C) and 37 compared to 75).

The tendency to treat the skin lesion increased by switching from instrument F (where dermatologists 1, 2, 3, and 4, respectively, indicated excision for 30, 56, 31, and 44 lesions) to instrument N (respectively, 36, 84, 75, and 57 lesions), and increased further with instrument V (respectively, 50, 99, 79, and 48 treated lesions). Furthermore, it was noted that dermatologist 2 generally had a greater tendency to remove lesions compared to dermatologists 1, 3, and 4.

### 3.1. Primary Outcome: Agreement between Dermatologists and Real Clinical Condition

The primary outcomes were: (i) to determine the concordance between the real clinical condition and the dermatologists’ judgments on the images acquired with the three instruments and (ii) to evaluate whether the mobile instruments (N and V) allowed a reliable remote diagnosis, thus exploiting TDS. The Cohen’s kappa index (K) was calculated and represented the agreement degree between the two methods, with the respective 95% CI (Table 2).

Most dermatologists, with respect to the instrument considered, had a K ≥ 0.61 (suboptimal agreement with K between 0.61 and 0.80). As regards the lower K values, we found a minimum value of 0.37, therefore the degree of agreement was never lower than “fair” (corresponding to K values between 0.21 and 0.40).

The agreement degree can also be expressed as a percentage (Table 3, part A), calculated as the ratio between the concordant values and the number of lesions. This value will always be greater than Cohen’s K, as it also includes the number of answers accidentally corrected. For example, the lower K value found (0.37) corresponds to an agreement of 64.58%, while the best value of K (0.87) indicates the agreement (94.44%) obtained by dermatologist 4 between instrument F and the real clinical condition.

### 3.2. Secondary Outcomes: Agreement between Dermatologists and Serious Diagnostic Error

Analysis of the agreement derived from the calculation of the Fleiss kappa (95% CI) for each instrument and each variable considered (diagnosis and treatment).

The agreement was at least moderate (K = 0.41) as regards the type of lesion diagnosed. In particular, an agreement degree was reached with K between 0.55 and 0.67; while for the treatment it was suboptimal for instrument F (K = 0.61), dropped to moderate (K0.44) for the Nurugo^TM^ (N), and became fair (K = 0.38) with the Nurugo^TM^ equipped with the slide to obtain epiluminescence images (V).

Then, we calculated the number of misdiagnosed malignant lesions (MM and BCC). In particular, we evaluated how many of these were classified as MN or SK and, therefore, assimilated to benign lesions; this corresponds to the number of patients in which the lesions should be surgically removed but in which, without a correct diagnosis, only a follow-up visit was planned. This misclassification has been defined as a “serious diagnostic error” and is shown in Table 3, part B (both absolute number and the percentage calculated on the total number of malignant lesions in the study [6 melanomas + 27 BCCs]).

For all the dermatologists involved, there were fewer serious diagnostic errors with the V instrument (maximum of four misclassified lesions) than with the F and N instruments (nine lesions misclassified); dermatologists 1 and 3 had an error rate that increased moving from instrument V (two errors) to instrument N (six and five errors, respectively), and increased even more to F (eight and nine misclassifications, respectively), whereas for dermatologist 2, we did not observe substantial differences based on the different instruments utilized (3–4 errors). Instead, dermatologist 4 had comparable results for instruments V and F (three–four lesions misclassified) and showed more errors (eight lesions misclassified) with N. This highlights the importance of exploiting the Nurugo Derma^TM^ to obtain epiluminescence images (V). Nevertheless, the total amount of misclassified lesions was limited, with a maximum of nine malignant lesions incorrectly classified (27.27%).

## 4. Discussion

The main purpose of this work is to evaluate whether the Nurugo Derma^TM^ inexpensive smartphone microscope could be potentially considered for the exchange of dermatological images between GP and specialist, to carry out a first “remote” patient screening.

The comparison between the images obtained with the Nurugo Derma^TM^ in two different settings and with conventional dermoscopy confirmed that the latter gave excellent agreement values (0.64 < K < 0.80) concerning the clinical reality (C) for all dermatologists. Instrument N gave good results but with average agreement values lower than the conventional dermoscopy (0.59 < K < 0.78), whereas the V instrument obtained a medium/moderate agreement, lower than the two previous instruments (0.53 < K < 0.66). The interpretation of these results concerns the different dermoscopical skills of the four dermatologists: with instrument F, the more experienced dermatologist 4 obtained “almost perfect” results (K = 0.85). However, the trend tended to reverse for N and V, for which dermatologists 1, 2, and 3 obtained higher agreement degrees. Indeed, TD, and more specifically, TDS, have developed recently and even a less experienced eye could be more versatile in adapting to the new required standards. Both for instrument F (K = 0.87) and instruments N and V, the same trend was recorded for the treatment indication. In addition, this result can be attributed to the high experience of dermatologist 4, who could identify, despite the artefacts, the lesion to be treated.

The secondary outcome of our study was to establish the agreement between the four dermatologists concerning diagnosis and treatment indication of the lesions evaluated through the three instruments. The degree of the diagnostic agreement was moderate/suboptimal, with better K values for F (K = 0.67), intermediate values for N (K = 0.63), and slightly lower values for V (K = 0.55). The excision indication reached a suboptimal agreement (K = 0.61) with F, while the N and V recorded moderate/intermediate agreement (respectively, K = 0.44 and K = 0.38). These results indicate that all the instruments can be potentially used for a first dermatological screening, while for the final treatment decision, the gold standard remains the dermatoscope.

Finally, the number of misdiagnosed malignant lesions (defined as “serious diagnostic errors”) was evaluated. We have shown that even the V instrument is effective in the first screening, with a limited number of errors (maximum 4/33 lesions). Moreover, dermatologist 4 demonstrated the importance of epiluminescence for a correct characterization of the malignant lesions (eight errors for N against three and four, respectively, for V and F).

Despite the novelty of this topic, we identified some already published similar studies with which to compare our results. Jesse et al. [16] evaluated the agreement between a multidisciplinary team in the analysis of images using store-and-forward teledermatology. In this study, the grade of agreement of the teledermatologist was moderate (K = 0.49) with the dermatologist and slightly lower (K = 0.40) with a primary care team. The gap widened when considering treatment, with a poor agreement reached by the hospital team (K = 0.12 for topical and K = 0.19 for systemic treatment). Our results are comparable to those obtained in this study by the dermatologist (K > 0.55 and K = 0.49, respectively); regarding the treatment, the final level of agreement was medium/moderate (K between 0.38 and 0.47 in both studies).

In the study by Kroemer et al. [17], images from 113 skin lesions were acquired through a smartphone equipped with a dermatoscopic lens, examined by a TDS-expert dermatologist, classified according to four diagnostic categories, and compared to the clinic-pathological diagnosis. An almost perfect agreement degree emerged, with K = 0.84. Additionally, this report is in line with our results, especially with the F instrument.

The aim of the study by Nami et al. [18] was to evaluate the efficacy and reliability of a web application system developed for mobile phones, comparing the classic face-to-face dermatological exam and the store-and-forward TD. A further objective was to investigate whether there was a real saving of time compared to a traditional visit. A total of 982 images of non-pigmented skin lesions were acquired by GPs using a smartphone equipped with an app facilitating the acquisition of digital images, patient data collection, and transmission to a secure website. Six dermatologists with similar experience performed the face-to-face exam, while an expert teledermatologist assessed all the cases. The study confirmed an almost perfect agreement (K = 0.906) between face-to-face and TD diagnosis, both for benign and malignant lesions, even if two skin tumors were misdiagnosed by the teledermatologist. In addition, this study highlighted the absolute importance of the dermatoscopy image for the correct identification of skin neoplasms. Even in our study, the misclassification of malignant lesions was reduced by using device V to acquire epiluminescence images (maximum 3/33 errors compared to a maximum of 8/33 for instrument N). Differences in the agreement grade regarding the treatment in comparison to our results can be explained by the higher sample size and the superiority of the camera used in this study [18]. This study [18] also underlined that the duration of the teleconsultation had an average time 25% higher than a face-to-face visit (19 min vs. 15 min), but there is a reduction in waiting times and in those required to reach the hospital, with a substantial time saving for the patient.

Finally, a Norwegian pilot study [19], tested an app (Askin^®^) that allowed GPs to write a short patient medical history and take clinical and dermatoscopic photographs of neoplastic and non-neoplastic skin lesions, recorded using a dermoscopy lens attached to the smartphone camera. All consultations were screened by a dermatologist; face-to-face specialist consultation was avoided for 70% of patients. According to our study results, TDS may be part of a triage system in which patients presenting suspicious skin lesions can be referred to the specialist through priority classes.

Despite the fact that artificial-intelligence-based algorithms have previously been demonstrated to outperform human experts in the diagnosis of the pigmented skin lesions [20], we strongly believe that clinical data could help increase the sensibility of human-guided teledermoscopy, and this was recently confirmed by Blum et al. [21]. On the contrary, given that in our study the same lesion was acquired with different tools (F, N, and V), in order to avoid image identification, the dermatologist did not have access to any clinical and/or anamnestic information related to the dermoscopy image.

In conclusion, our study demonstrates that the Nurugo^TM^ Derma, an inexpensive (~$50) device created for amateur use, can represent a useful tool for a first-level dermatological and dermatoscopy screening, and with a specific support network, can allow the GP to carry out a preliminary screening of the patients to be referred to an expert dermatologist. This has considerable advantages in terms of rationalizing resources and reducing potential inconvenience for patients, especially in the current pandemic context.

The potential applications of this device in the dermatological field are numerous and its use could be extended to other skin disease diagnoses and follow-ups, such as chronic inflammatory diseases, especially when face-to-face visits are difficult. Therefore, Nurugo^TM^ Derma could be exploited also by fragile, immunosuppressed, or disabled patients, or in emergency contexts in which access to the hospital is limited to a non-deferrable situation. Furthermore, such a tool could also be used in the dialogue between dermatologists, to get a quick opinion from a colleague about a lesion of a dubious nature. However, adequate training for use of the device is fundamental in order to acquire good-quality images. In our experience, despite some artifacts connected to the instrument (poor focus, presence of air bubbles or streaks, and reduced FOV), the obtained images can adequately show the characteristics of the skin lesions and the dermoscopic patterns necessary for an accurate diagnosis. Therefore, by developing adequate correction and normalization algorithms of the acquired images, the Nurugo^TM^ Derma^®^ can represent a useful and valid tool in clinical practice.

## Figures and Tables

**Figure 1 diagnostics-12-01371-f001:**
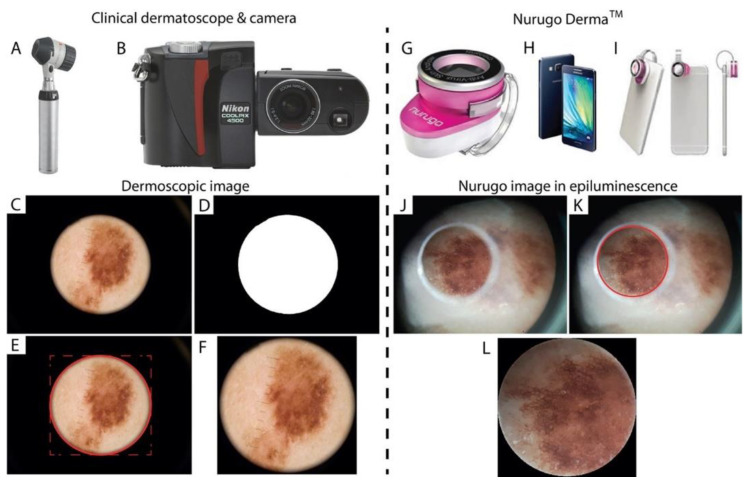
(**Top**) Representation of the portable dermatoscope connected to the camera (**A**,**B**) and of the Nurugo^TM^ Derma (**G**) connected to the smartphone (**H**,**I**); (**Bottom**) Examples of images acquired with the different devices and image pre-processing (Dermatoscope from (**C**–**F**); Nurugo^TM^ Derma from (**J**–**L**)).

**Figure 2 diagnostics-12-01371-f002:**
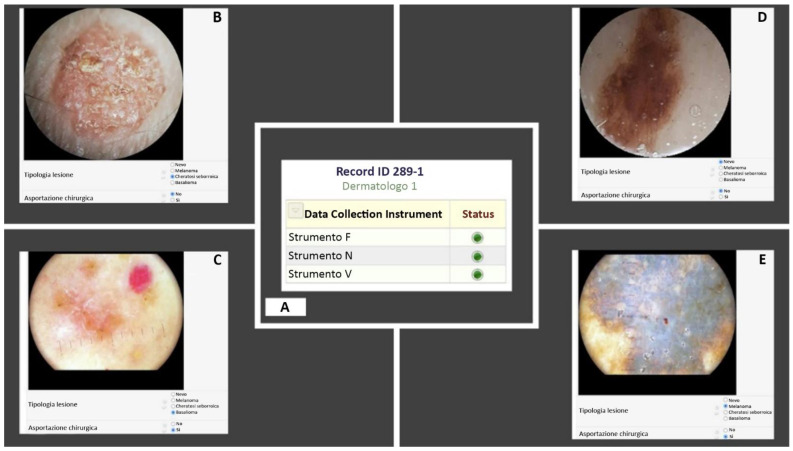
Examples of images to be evaluated through the REDCap software; (**A**) Table of the instruments in use. The images were acquired with instrument N (**B**), F (**C**) and V (**D**,**E**).

**Table 1 diagnostics-12-01371-t001:** Results of the descriptive analysis separated for the three different instruments and for the four dermatologists. Absolute and % values are reported. Instrument F: dermatoscope; Instrument N: Nurugo; Instrument V: Nurugo in epiluminescence setting with slide; C: real clinical condition.

	Dermatologist 1	Dermatologist 2	Dermatologist 3	Dermatologist 4	Real Clinical Diagnosis C
**LESION TYPE**					
Instrument F					
Nevus	85 (59.03%)	64 (44.44%)	73 (50.69%)	72 (50%)	75 (52.08%)
Melanoma	10 (6.94%)	18 (12.5%)	9 (6.25%)	3 (2.08%)	**6** (4.17%)
Seborrheic keratoses	29 (20.14%)	39 (27.08%)	44 (30.56%)	41 (28.47%)	36 (25.00%)
Basal cell carcinoma	20 (13.89%)	23 (15.97%)	18 (12.5%)	28 (19.44%)	27 (18.75%)
Instrument N					
Nevus	72 (50%)	68 (47.22%)	61 (42.36%)	46 (31.94%)	
Melanoma	**7** (4.86%)	**16** (11.11%)	**24** (16.67%)	**3** (2.08%)	
Seborrheic keratoses	36 (25%)	35 (24.31%)	36 (25%)	57 (39.58%)	
Basal cell carcinoma	29 (20.14%)	25 (17.36%)	23 (15.97%)	38 (26.39%)	
Instrument V					
Nevus	76 (52.78%)	52 (36.11%)	70 (48.61%)	**37** (25.69%)	
Melanoma	**17** (11.81%)	**29** (20.14%)	**20** (13.89%)	**9** (6.25%)	
Seborrheic keratoses	23 (15.97%)	44 (30.56%)	29 (20.14%)	**64** (44.44%)	
Basal cell carcinoma	28 (19.44%)	19 (13.19%)	25 (17.36%)	34 (23.61%)	
**TREATMENT**					
Instrument F					
SI	30 (20.83%)	**56** (38.89%)	31 (21.53%)	44 (30.56%)	48 (33.33%)
Instrument N					
SI	36 (25.00%)	**84** (58.33%)	75 (52.08%)	57 (39.58%)	
Instrument V					
SI	50 (34.72%)	**99** (68.75%)	79 (54.86%)	48 (33.33%)	

**Table 2 diagnostics-12-01371-t002:** Cohen’s K values with respective 95% CI.

	Dermatologist 1	Dermatologist 2	Dermatologist 3	Dermatologist 4
**LESION TYPE**				
F vs. C	0.64 [0.53–0.75]	0.80 [0.72–0.88]	0.70 [0.60–0.80]	0.85 [0.77–0.92]
N vs. C	0.75 [0.66–0.84]	0.78 [0.69–0.86]	0.62 [0.51–0.71]	0.59 [0.49–0.69]
V vs. C	0.65 [0.55–0.75]	0.60 [0.51–0.70]	0.66 [0.56–0.76]	0.53 [0.42–0.64]
**TREATMENT**				
F vs. C	0.69 [0.56–0.82]	0.79 [0.69–0.89]	0.67 [0.54–0.80]	0.87 [0.79–0.96]
N vs. C	0.63 [0.50–0.77]	0.42 [0.29–0.55]	0.38 [0.24–0.52]	0.57 [0.43–0.71]
V vs. C	0.60 [0.46–0.74]	**0.37 [0.26–0.48]**	0.50 [0.38–0.63]	0.63 [0.49–0.76]

**Table 3 diagnostics-12-01371-t003:** **Part A.** Percentage agreement (%). **Part B**. Absolute number and relative percentage of “serious diagnostic errors”.

Part A	Dermatologist 1	Dermatologist 2	Dermatologist 3	Dermatologist 4
**LESION TYPE**				
F vs. C	77.78%	86.81%	81.25%	90.28%
N vs. C	84.03%	85.42%	73.61%	72.22%
V vs. C	77.78%	72.22%	77.78%	67.36%
**TREATMENT**				
F vs. C	87.50%	90.28%	86.81%	94.44%
N vs. C	84.72%	69.44%	68.75%	79.86%
V vs. C	81.94%	64.58%	74.31%	83.33%
**Part B**	Dermatologist 1	Dermatologist 2	Dermatologist 3	Dermatologist 4
**LESION TYPE**				
F vs. C	8 (24.24%)	3 (9.09%)	9 (27.27%)	4 (12.12%)
N vs. C	6 (18.28%)	3 (9.09%)	5 (15.15%)	8 (24.24%)
V vs. C	2 (6.06%)	4 (12.12%)	2 (6.06%)	3 (9.09%)

## Data Availability

The data presented in this study are available on request from the corresponding author.

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
