# Peer review of "Teledermoscopy in the Diagnosis of Melanocytic and Non-Melanocytic Skin Lesions: NurugoTM Derma Smartphone Microscope as a Possible New Tool in Daily Clinical Practice"

_diagnostics, 2022, doi:10.3390/diagnostics12061371_

Round 1

Reviewer 1 Report

This is a very interesting article.  With a good training in photograph interpretation  lot of people with unaccesible health services  will recieve help and early treatment in case of malignancy

What about costs of the appliance? some times the mobile accesorias as de dermoscope is more expensive than the Dermlite.

Author Response

Reviewer 1.

This is a very interesting article.  With a good training in photograph interpretation lot of people with unaccesible health services will receive help and early treatment in case of malignancy

What about costs of the appliance? some times the mobile accessories as de dermoscope is more expensive than the Dermlite.

Authors’ reply: Thank you very much for your comment. We agree that teledermoscopy could represent a valid option allowing precocious diagnosis and treatment, also when the access to the hospital is limited as during the recent pandemic. The Nurugo Derma device is ready-to-use, easy to buy and obtain (easily available on the most popular websites) and its cost (about 50 €) is affordable. Moreover, it is definitely cheaper than the dermoscopes available in the dermatology practice, and the smartphone Dermlite devices DermLite HUD and DermLite handyscope have a respective cost of 155€ (plus the potential cost of a smartphone adapter) and $795 on their website.

In order to highlight the low cost of the Nurugo device, we added once more the term “inexpensive (~$50)” at line 313:

“In conclusion, our study demonstrates that the NurugoTM Derma, an inexpensive (~$50) device created for amateur use”

Reviewer 2 Report

The paper is well designed and shows promising results for the use of the suggested technology. However, the missing user behavioural questionnaire is somehow limiting the non-professional or professional use of the detection tool. The additional questionnaire may increase the potential of the technology by adding user experience with this microscopy. Technological improvement should be discussed for this purpose 

Author Response

Reviewer 2.

The paper is well designed and shows promising results for the use of the suggested technology. However, the missing user behavioral questionnaire is somehow limiting the non-professional or professional use of the detection tool. The additional questionnaire may increase the potential of the technology by adding user experience with this microscopy. Technological improvement should be discussed for this purpose 

Authors’ reply: We thank you for the comment. The principle aim of present study is to evaluate skin lesions through the Nurugo Derma device compared with the traditional dermoscope and its usefulness in a first level screening, as it may occur in the General Practitioner’s (GP) Outpatients Clinic. With this in mind, we do not assume that the general population will use the Nurugo Derma device, but that its use will still be limited within the medical field. Hence, the behavioural questionnaire is not appropriate, since the GP is expected only to acquire the image and transfer it to the dermatologist through Teledermoscopy. Regarding the dermatologist experience with both the Nurugo Derma and the traditional dermoscope, as already stated in the manuscript, the 4 specialists had different grade of experience (see lines 131-132 and 239-248). 

In the literature, the technological improvement of teledermoscopy and artificial intelligence have been previously discussed, also when considering the use of smartphone applications (Apps). In such setting, it has been already highlighted the need for clinical data concerning the image, since they could be of help in formulating the diagnosis.

On the contrary, since in our study each lesion had to be classified in one of the four established categories (melanoma, melanocytic nevus, basal cell carcinoma and seborrheic keratosis), and in order to avoid an easy recognition of the image, we decided to not add any additional clinical and anamnestic information in the REDCap records (as specified in the revised version, lines 125-126). Accordingly, since the dermatologists were asked to formulate a clinical diagnosis and also the possible indication for surgical removal on the basis of the sole dermoscopy image, we can assume that this could have a negative impact in the evaluations and that the excision could be suggested for a higher number of lesions. For all these reasons, we believe that the behavioral questionnaires are not appropriate in this current study.

In order to discuss such points, we edited the discussion (lines 306-312) and we added a couple of references (20 and 21):

“Despite the fact that artificial intelligence-based algorithms have previously demonstrated to outperform human experts in the diagnosis of the pigmented skin lesions [20], we strongly believe that clinical data could help increase the sensibility of hu-man-guided teledermoscopy, and this was recently confirmed by Blum et al [21]. On the contrary, since in our study the same lesion was acquired with different tools (F, N and V), in order to avoid image identification, the dermatologist did not have access to any clinical and/or anamnestic information related to the dermoscopy image.”